# Energetic [1,2,5]oxadiazolo [2,3-*a*]pyrimidin-8-ium Perchlorates: Synthesis and Characterization

**DOI:** 10.3390/molecules27238443

**Published:** 2022-12-02

**Authors:** Kirill V. Strizhenko, Anastasia D. Smirnova, Sergei A. Filatov, Valery P. Sinditskii, Adam I. Stash, Kyrill Yu. Suponitsky, Konstantin A. Monogarov, Vitaly G. Kiselev, Aleksei B. Sheremetev

**Affiliations:** 1Zelinsky Institute of Organic Chemistry, Russian Academy of Sciences, 119991 Moscow, Russia; 2Mendeleev University of Chemical Technology, 125047 Moscow, Russia; 3Nesmeyanov Institute of Organoelement Compounds, Russian Academy of Sciences, 119334 Moscow, Russia; 4Semenov Federal Research Center for Chemical Physics, Russian Academy of Sciences, 119991 Moscow, Russia; 5Institute of Chemical Kinetics and Combustion, Russian Academy of Sciences, Siberian Branch, 630090 Novosibirsk, Russia

**Keywords:** aminofurazans, fused furazans, perchlorates, energetic compound, synthesis, X-ray analysis, impact sensitivity, thermal decomposition, combustion, burning rate

## Abstract

A convenient method to access the above perchlorates has been developed, based on the cyclocondensation of 3-aminofurazans with 1,3-diketones in the presence of HClO_4_. All compounds were fully characterized by multinuclear NMR spectroscopy and X-ray crystal structure determinations. Initial safety testing (impact and friction sensitivity) and thermal stability measurements (DSC/DTA) were also carried out. Energetic performance was calculated by using the PILEM code based on calculated enthalpies of formation and experimental densities at r.t. These salts exhibit excellent burn rates and combustion behavior and are promising ingredients for energetic materials.

## 1. Introduction

Most energetic materials are a mixture of substances, among which an oxidizer, fuel, binder and various corrective additives can be specified. The most widely used oxidizer is ammonium perchlorate, NH_4_ClO_4_ [1,2,3,4]. In the early stages of research, compositions based on NH_4_ClO_4_ included powdered metals as fuel. However, in recent years, there has been a tendency to partially or even completely replace metallic fuels in energetic materials with high-nitrogen compounds characterized by high positive enthalpies of formation.

Energetic salts with nitrogen-rich organic cations and/or anions (typically azole-based ions) are a major area for the development of high-energy materials, since salts have a high density and positive enthalpy of formation, are often thermally stable, and are not volatile [5,6,7]. Designing energetic salts by combining various cations and anions to achieve a specific purpose is a simple but powerful methodology. Most of these salts, however, have a strongly negative oxygen balance and, with a lack of oxidizer, form toxic or undesirable solid decomposition products [8].

When using energetic salts in combination with solid inorganic oxidizers—for example, with NH_4_ClO_4_—additional problems are observed. A metathesis reaction between NH_4_ClO_4_ and energetic azole-based salts leads to the formation of new salts where the anions have been swapped. As a result, the properties of this composition could change unpredictably [9]. To overcome this disadvantage, a variety of energetic organic salts may be limited by the use of perchlorates.

The characteristics of the energetic salts are dictated by the physical and chemical features of both ions in their composition. In this light, the importance of synthesizing organic salts of perchloric acid with various cations and the study of their properties becomes important. It is important to note that, unlike nitrates, perchlorates have a significant effect on the transformation of organic compounds [10,11,12], which favors the completeness of the combustion of the latter.

Due to its inherent density, positive enthalpy of formation, thermal stability, and the presence of an active oxygen atom, the 1,2,5-oxadiazole (furazan) ring is an attractive building block for the development of new energetic compounds [13,14,15,16,17]. Significant progress has been achieved in the development of furazan-based salts, some types of which are depicted in Figure 1. Typically, the furazan moiety is located in the anionic part of the energetic salt, as in salts of nitramines **1** [18,19,20,21,22,23], perchlorylamines **2** [24], dinitromethyl **3** [25,26,27,28,29,30,31,32], tetrazolyl **4** [33,34,35,36,37], pyrazolo [3,4-*c*]furazanates **5** [38,39,40], and [1,2,3]triazolo [4,5-*c*][1,2,5]oxadiazoles **6** [41]. Cations involving the furazan ring are very rare; in the previously described salts (**7 [42]** and **8 [20]**), the positive charge is located in the side chain.

There is no literature precedence for energetic salts incorporating a furazan-based backbone with a positive charge on the nitrogen atom of this ring.

Although a few 2-methyl-1,2,5-oxadiazolium [43] and 1,2,5-oxadiazolo [2,3-*a*]-pyrimidinium perchlorates [44,45] were synthesized, only partial physical and spectral properties were available, but the sensitivity, thermal stability, combustion features, and explosive performance were not reported. However, these early studies have demonstrated that monocyclic salts are less accessible and more reactive.

In light of the above, we report here a new, straightforward method for the synthesis of energetic 1,2,5-oxadiazolo [2,3-*a*]-pyrimidinium perchlorates, and their full characterization, including energetic properties. The only known pathway for the synthesis of 1,2,5-oxadiazolo [2,3-*a*]-pyrimidinium perchlorates is based on the reaction of 3-amino-4-R-furazans **9** with 1,3-dicarbonyl compounds [44]. However, an investigation of the scope and limitations of functionalized reactants has not been previously reported. Although a range of alkyl and aryl-substituted oxadiazolo [2,3-*a*]-pyrimidinium perchlorates have been prepared by this approach, there have been no examples bearing explosophoric groups [46] so far.

## 2. Results

### 2.1. Synthesis

Readily available 3-amino-4-methylfurazan (**9a**) [47] became our model precursor of choice. There is a literature report [44] on the synthesis of 1,2,5-oxadiazolo [2,3-*a*]-pyrimidinium perchlorate **10a** from compound **9a** and pentane-2,4-dione in a mixture of AcOH and HClO_4_, but no synthetic details are given. In our hands, such a procedure turned out to be somewhat unpredictable; the yield of the final 1,2,5-oxadiazolo [2,3-*a*]-pyrimidinium perchlorate **10a** was relatively low (only 23% vs. 73% reported earlier).

After several experiments, we found that replacing AcOH in the reaction mixture with Ac_2_O improved the conversion significantly and gave a 44% yield of the product **10a** (Figure 1). A change in dehydrating agent from Ac_2_O to (CF_3_CO)_2_O allowed us to increase the yield up to 87%. It has been suggested that the solubility of the product **10** in the reaction mixture is the most important in this process. (CF_3_CO)_2_O is completely unnecessary as a reagent, and the much cheaper and easier-to-handle trifluoroacetic acid was employed instead. Gratifyingly, treatment of amine **9a** with pentane-2,4-dione in a mixture of 58% HClO_4_ and CF_3_CO_2_H at room temperature gave the cyclocondensation product **10a** in 3 h. Thus, without any complicated workup, the desired product was isolated in 71% yield and >99% purity through simple filtration and washing.

With the optimized conditions developed, we explored the scope of the method with a variety of 1,3-dicarbonyl compounds and aminofurazans bearing explosophoric groups. A 1,3-dicarbonyl compound with one halogen, namely 3-chloropentane-2,4-dione, furnished the desired product **10b** in a good yield (85%). The use of 1,3-dicarbonyl compounds possessing strongly electron-withdrawing substituents, e.g., trifluoromethyl or nitro groups (Figure 1), failed to give the corresponding bicycles **10c** and **10d**.

Figure 2 shows the results of applying our optimized cyclocondensation conditions to a range of aminofurazans. Aminofurazans bearing a functionalized methyl group, 3-amino-4-chloromethylfurazan (**9b**) [48] and 3-amino-4-azidomethylfurazan (**9c**) [48], were equally effective as the model compound **9a**, with excellent yields (ca. 95%) being obtained for pentane-2,4-dione in the HClO_4_/CF_3_CO_2_H system. Moreover, 3-amino-4-azidofurazan (**9d**) [49,50] was a good substrate for the reaction and gave the target product **9e** in 71% yield. Synthesis of 1,2,5-oxadiazolo [2,3-*a*]-pyrimidinium perchlorate failed in cases where the starting furazan had a very strong electron-withdrawing substituent: reactions with 3-amino-4-nitrofurazan **9e** [51] or with 3-amino-4-*tert*butylazoxyfurazan **9f [52]** did not give the desired salts **10h** and **10i**. Only recovery of the starting materials from the reaction mixture was achieved.

With the exception of compound **10a**, all perchlorates prepared in this way are new, and the structures of the products have been confirmed by elemental analyses, IR, and ^1^H, ^13^C, ^15^N, and ^35^Cl NMR spectroscopy. The ^1^H, ^13^C and ^15^N chemical shift assignments were determined by using ^1^H selective NOE, ^1^H-^13^C HSQC, ^1^H-^13^C HMBC, and ^1^H-^15^N HMBC experiments. In the ^35^Cl NMR spectrum, the characteristic chlorine signal of the perchlorate anion appeared at 1012 ppm, which is close to the reported values of similar compounds [53]. For clarity, the compound of this study, together with relevant NMR parameters, is depicted in Figure 2.

### 2.2. X-ray Analysis

The X-ray crystal structures of perchlorates **10a**, **10f**, and **10g**, differing only by the substituent at the 1,2,5-oxadiazole ring, are shown in Figure 3 and Figure 4. Both symmetrically independent molecules of methyl compound **10a** adopt a planar structure. Azido groups in three symmetrically independent molecules of **10g** are rotated ca. 17.6°–26.7° out of the plane of the bicyclic backbone. For compound **10f**, the CH_2_N_3_ substituent deviates even more significantly from the plane of the heterocycle (the C2–C1–C8–N4 torsion angle is equal to 59.7(2)°).

The N–O bond lengths in the 1,2,5-oxadiazole ring, which are usually the most sensitive to the influence of substituents [54,55,56], are distributed so that the O1–N2 bond is significantly shorter than the other one in all three compounds. The difference in these bonds Δ(NO) (for **10a** and **10g**, the average value over symmetrically independent molecules is used) increases in the order of the decreasing electron-withdrawing effect of the substituent at the C1 atom (the Δ(NO) is 0.022, 0.026, 0.035 Å for compounds **10g**, **10f**, and **10a**, respectively).

For all three compounds, the molecules in the crystal are linked together by O…π, C–H…O(N), and van-der-Waals interactions. As expected, most of them are observed between anions and cations, as depicted in Figure 5, Figure 6 and Figure 7. In the cases of azido compounds (Figure 6 and Figure 7), each anion (except for one in **10g**, Figure 6) is linked to four cations by means of O…π and C–H…O interactions, and the interaction patterns are quite similar for both compounds. In the case of Me derivatives (Figure 5), each anion is surrounded by three counterions. In all three structures, the cation…cation interactions are caused by van-der-Waals forces and weak C–H…N hydrogen bonds, while no anion…anion contacts are observed.

As expected, the density of Me compound **10a** (1.549 g cm^−3^ at 100 K) is lower than that of its azido analogues. However, unexpectedly, the density of the azido derivative **10g** turned out to be lower than that of the azidomethyl compound **10f** (1.611 vs. 1.649 g cm^−3^ at 100 K). Probably due to the significant disorder in the structure **10f**, disordered fragments occupy a larger volume, which leads to a decrease in density.

### 2.3. Initial Safety Testing

For initial safety testing, the impact (IS) and friction (FS) sensitivities of perchlorates **10a**, **10e**–**10g** were measured. The sensitivity measurements were carried out using common BAM techniques and compared to tetrazene, which is considered as the benchmark primary metal-free explosive (Table 1). Compound **10a** bearing the methyl group at 1,2,5-oxadiazole ring and compound **10e** with the chloromethyl group showed similar impact and friction sensitivities. A change in the position of the chlorine atom—namely, its transfer from the methyl group to the pyrimidine ring, as in compound **10c**—slightly reduces the sensitivity to friction and increases the thermal stability. Unexpectedly, when the chloromethyl group was replaced with an azidomethyl group, as in compound **10f**, the sensitivity decreased to 2.6 J. The friction sensitivity of compound **10a** is two times lower than that of the chlorine derivative **10e**, and two times higher than that of azidomethyl compound **10f**. The primary differential scanning calorimetric (DSC, 5 °C min^−1^) tests are also presented in Table 1.

All perchlorates in this study are more thermally stable than tetrazene, slightly exceeding it in impact sensitivity, but less sensitive to friction. Compound **10g** with the azido group at the 1,2,5-oxadiazole ring is the most impact- and friction-sensitive in this series of salts.

### 2.4. Thermal Analysis

Since the azide compounds **10f** and **10g** are the most energetic, we continued to characterize them and the model compound **10a** from the point of view of their decomposition under heating. The thermal stabilities of the compounds were determined by DSC and thermogravimetric analysis (TGA) measurements scanning at 10 °C min^−1^. The model compound **10a** decomposes without melting, and the maximum heat release was observed at 218–219 °C, which is comparable to that of common RDX. However, decomposition proceeds very intensively; an acceptable DSC curve can only be obtained on a sample less than 0.2 mg. In TGA measurements, the weight loss occurring at this temperature is approximately 19%, which is close to MeCN release (15.6%), as previously observed for 1,2,5-oxadiazoles [48,57,58,59,60,61,62]. Further weight loss is observed after 280 °C, which characterizes the second stage of decomposition (see Appendix A).

Decomposition of the model compound **10a** under isothermal conditions was performed using a Bourdon glass compensation pressure gauge [63] in the temperature range of 160–180 °C. The ratio of the weight of the sample to the volume of the reaction vessel (m/V) was ~1 × 10^−3^ g cm^3^. The destruction of **10a** under this condition proceeds with an acceleration in time, which, at temperatures above 160 °C, turns into a degenerate thermal explosion (Appendix A). During decomposition, only 60–95 cm^3^ g or 0.7–1.11 moles of colorless gaseous products were released from a mole of the initial **10a**, most of which condenses during cooling. After stopping the process, a dark brown powder remains at the bottom of the vessel. An absorption band of the ClO_4_^−^ (1100 cm^−1^) anion was observed in the IR spectrum of the residue.

A probable mechanism of the initial destruction stage of salt **10a** is proposed in Figure 3. This is consistent with the typical decomposition mechanism of 1,2,5-oxadiazoles previously observed for various compounds of this heterocycle [57,58,60,62]. The breaking of the two bonds of the 1,2,5-oxadiazole ring (indicated in the Figure 3) gives the nitrile product **A** as a result of the typical ring disintegration process. Rapid intermolecular cyclization of the resulting reactive monocyclic intermediate **B** leads to a tricyclic intermediate **C**, capable of further transformations.

X-ray diffraction analysis of **10a** shows that the O1–N2 bond is significantly shorter than the other O1–N1 bond of the 1,2,5-oxadiazole ring, which explains the breaking of the latter. The IR spectrum of the final residue includes bands corresponding to C=N bonds (1640, 1590, 1430 cm^−1^), as well as a strong absorption band of the ClO_4_^−^ anion (1100 cm^−1^).

The main heat effect is achieved due to the formation of C–O bonds during the formation of tricyclic compound **C**. The formation of two new C–O bonds, even without taking into account the change in the enthalpy part, makes it possible to estimate the heat effect at 746 J g^−1^, which is in good agreement with the DSC data (1164 J g^−1^).

Compound **10f** bearing the CH_2_N_3_ group at the 1,2,5-oxadiazole ring melts at 142 °C (enthalpy of melting L_m_ > 23 J g^−1^), and, immediately after this, the exothermic stage of decomposition begins (Appendix A). Decomposition of **10f** proceeds in two stages. At the first, occurring at 143–180 °C with a maximum at 159 °C (1438 J g^−1^), the weight reduction was 27.6%, which corresponds to a loss of N_3_CH_2_CN (26.9%), as a result of the breaking of two bonds in the 1,2,5-oxadiazole ring, similar to what was observed for **10a** (Figure 3). With such isothermal decomposition of **10f**, the gas release is 120 cm^3^ g^−1^ or 1.66 mol per mol of the compound (see Appendix A). It is obvious that, in this case, there is also a decomposition of the azide group.

Compound **10g** bearing an electron-withdrawing azide group at the 1,2,5-oxadiazole ring is slightly less thermally stable. When heated using a 10 °C min^−1^ ramp rate, it decomposes in the range of 140–164 °C (maximum heat release of 150 °C). Total energy of the decomposition was 2061 J g^−1^. Weight loss at this stage reaches 54%, which indicates a deeper decomposition process than is shown in Figure 3. On the DSC curve, there is another area with weaker heat release at 325 °C, where the weight loss is 21% (see Appendix A).

The decomposition of all compounds of this study under isothermal conditions proceeds with an acceleration in time, and, after completion, a dark brown powder remains at the bottom of the vessel. Here, decomposition was carried out at temperatures below the onset decomposition temperature registered in the DSC. The observed acceleration is associated with the submelting of samples. The analysis of such gas release curves makes it possible to simultaneously determine two decomposition constants, (*i*) in the solid phase (k_s_) and (*ii*) in the liquid phase (k_liq_). The obtained data are summarized in Table 2 and Appendix A. The activation energies of decomposition in the solid phase are relatively low (145–164 kJ mol^−1^) and, within the measurement error (±10 kJ mol^−1^), practically do not change during the transition to the liquid phase (152–157 kJ mol^−1^). In general, there are two trends in the data in Table 2. The first is that the decomposition in the liquid phase is much faster than in the solid phase. The second trend is that the rate decomposition constants of the perchlorates of this study, both in the solid phase and in the melt phase, and the Hammett constants of the substituent at the 1,2,5-oxadiazole ring, increase synchronously. A good correlation between the rate of thermal decomposition and the Hammett constant indicates a unified mechanism of the initial stage of destruction of these compounds.

Since the decomposition of compounds proceeds with acceleration, the data obtained under non-isothermal conditions (see Appendix A) give the formal kinetics of the total process. Previously, the decomposition of tetrazene was described only under non-isothermal conditions [65]. Comparison of the decomposition kinetics under the same conditions (see Appendix A) reveals that the perchlorates of this study are superior to tetrazene in thermal stability. The decomposition rate constant of tetrazene at 150 °C is several orders of magnitude higher than that of the studied perchlorates (Table 2).

### 2.5. Combustion

The combustion behaviors of the perchlorates were studied on pressed charges in polyurethane tubes (4 mm inner diameter and ca. 8 mm length). However, at elevated pressures, combustion in the tubes turns into an explosion. For example, the compound **10g** could be burned only at a pressure below 3 MPa. For compound **10f**, a clear result was obtained only when using charges in the form of thin (~1 mm) plates pressed to a high density. The use of such charges avoids the penetration of hot gases into the pores and prevents the transition of layer-by-layer combustion in convective ones. As a result, compound **10f** was able to burn even at high pressures (Figure 8 and Table 3).

As can be seen from Figure 8, the burning rates of compounds **10** are significantly higher than HMX (1,3,5,7-tetranitro-1,3,5,7-tetraazacyclooctane) [66] and comparable to the burning rates of tetrazene [67].

Remarkably, the burning rates of the perchlorates of this study increase in the following order: **10f** < **10g** < **10a**. However, a similar trend is observed in terms of increasing stability; that is, the decomposition rate of these perchlorates decreases. This result demonstrates a drastic difference from the usual correlations, when an increase in thermal stability leads to a decrease in the burning rate. It can be assumed that the thermolysis of perchlorates, which proceeds with the formation of gaseous products, is a process that determines the temperature of their surfaces during combustion. In this case, the more stable the perchlorate, the higher the temperature of its surface during combustion. If the burning rate is determined by reactions in the condensed phase, which is typical for compounds with low thermal stability [68], the surface temperature at which the leading combustion reaction takes place will be a more significant factor than the decomposition rate.

### 2.6. Explosive Performance

To evaluate the performance of these newly synthesized compounds, the enthalpies of formation were calculated (see Appendix A) and are summarized in Table 4. Even for the model compound **10a** bearing three ballast methyl groups, the enthalpy of formation is positive and there is +0.56 kJ g^−1^, which is twice as high as that of benchmark 1,3,5-trinitro-1,3,5-triazinane (RDX; +0.32 kJ g^−1^). As expected, the introduction of an azide group into one methyl group, and, moreover, the replacement of the methyl group with an azide group, significantly increases the enthalpy of formation, which exceeds the value for tetrazene [69]. This is clearly seen in Table 4. It is obvious that the further replacement of the remaining methyl groups with explosophoric substituents will make it possible to design more effective target products.

With the crystal density and enthalpy of formation data in hand, the explosive performance of perchlorates was demonstrated with the refined empirical methods implemented in the PILEM code [70]. Since the salts in Table 4 are more than two times poorer in oxygen than RDX (α = 0.667), their detonation velocities are lower than those of RDX (*D* = 8850 m s^−1^), close to that of trinitrotoluene (TNT, *D* = 6663 m s^−1^), and slightly lower than that of tetrazene.

The results presented in Table 2 and Table 4 show that compounds **10f** and **10g** bearing CH_2_N_3_ and N_3_ groups, respectively, have similar performance with respect to detonation velocity and detonation pressure, whereas **10f** is less sensitive to impact and friction. This feature can be used for the further tuning of the [1,2,5]oxadiazolo [2,3-*a*]pyrimidin-8-ium backbone.

## 3. Materials and Methods

**Caution:** Although we have encountered no difficulties during preparation and handling of these products, they are potentially explosive energetic materials. Manipulations must be carried out by using appropriate standard safety precautions.

Most of the reagents and starting materials were purchased from commercial sources and used without additional purification. The starting 3-amino-4-methylfurazan (**9a**) [47], 3-amino-4-chloromethylfurazan (**9b**) [48], 3-amino-4-azidomethylfurazan (**9c**) [48], 3-amino-4-azidofurazan (**9d**) [49,50], 3-amino-4-nitrofurazan **9e** [51], and 3-amino-4-*tert*butylazoxyfurazan **9f** [52] were synthesized by using previously reported procedures.

IR spectra were recorded on a BrukerALPHA instrument in KBr pellets. The ^1^H and ^13^C, ^14^N spectra were recorded on a Bruker AM-300 instrument (300.13, 75.47, and 21.69 MHz, respectively) at 299 K. The chemical shifts of ^1^H and ^13^C nuclei were reported relative to TMS (^1^H and ^13^C, 0.00 ppm). ^1^H–^15^N HMBC experiments were run to measure the ^15^N chemical shifts (^15^N, relative to liquid NH_3_, 0.00 ppm). Elemental analysis was performed on a PerkinElmer 2400 Series II instrument. Analytical TLC was performed using commercially pre-coated silica gel plates (Kieselgel 60 F_254_), and visualization was effected with short-wavelength UV light.

Thermal stability was studied by differential scanning calorimetry (DSC) using a Mettler Toledo DSC 822e module. The sample (1–2 mg) was weighed in an aluminum crucible (40 µL), sealed under air with a press, and then pierced with a needle to leave two holes with a diameter of ca. 1 mm. The decomposition of a sample was carried out in a nitrogen atmosphere at a purge rate of 50 μL min^−1^. The temperature of the onset of intense decomposition (T_onset_) was taken as the temperature determining thermal stability. The samples were subjected to thermostating in the measuring cell at a temperature of 25 °C for 30 min before the start of measurements.

Impact and friction sensitivities were measured with a BAM-type apparatus in a series of experiments according to STANAG procedures [71,72].

The burning rate was determined in a constant-pressure device (Crawford bomb) with a volume of 2 L in a nitrogen atmosphere. The combustion process of the sample was recorded using a pressure strain gauge, which transmitted the signal to a digital oscilloscope. The start and end times of combustion were determined from oscillograms. The burning rate was calculated by dividing the sample height by the burning time and was related to the mean integral pressure during the experiment. The error in determining the burning rate did not exceed 3%.

**Single-Crystal X-ray Diffraction Study.** The single crystals of perchlorates **10a**, **10f**, and **10g** were grown by crystallization from hot AcOH solution. Single-crystal X-ray diffraction experiments were carried out using a SMART APEX2 CCD diffractometer (λ(Mo-Kα) = 0.71073 Å, graphite monochromator, ω-scans) at 100 K. Collected data were processed by the SAINT and SADABS programs incorporated into the APEX2 program package [73]. The structures were solved by the direct methods and refined by the full-matrix least-squares procedure against *F*^2^ in anisotropic approximation. The refinement was carried out with the SHELXTL program [74]. The CCDC numbers (2210557 for **10a**, 2210558 for **10f**, and 2210559 for **10g**) contain the supplementary crystallographic data for this paper. These data can be obtained free of charge via www.ccdc.cam.ac.uk/data_request/cif, accessed on 1 November 2022.

*Crystallographic data for compound***10a**: C_7_H_7_N_6_O^+^ClO_4_^−^ are monoclinic, space group *P*2_1_/*c*: *a* = 15.1208(6) Å, *b* = 11.7428(5) Å, *c* = 20.4811(8) Å, *β* = 98.7126(13)°, *V* = 3594.7(3) Å^3^, *Z* = 12, M = 290.64, *d*_cryst_ = 1.611 g·cm^−3^. *wR*2 = 0.2394 calculated on *F*^2^*_hkl_* for all 7199 independent reflections with 2*θ* < 52.5° (*GOF* = 1.004, *R* = 0.0680 calculated on *F_hkl_* for 5235 reflections with *I* > 2σ(*I*)).

*Crystallographic data for compound***10f**: C_8_H_9_N_6_O^+^ClO_4_^−^ are monoclinic, space group *P*2_1_/*n*: *a* = 8.1920(3) Å, *b* = 13.8353(5) Å, *c* = 10.8329(4) Å, *β* = 91.751(2)°, *V* = 1227.21(8) Å^3^, *Z* = 4, M = 304.66, *d*_cryst_ = 1.649 g·cm^−3^. *wR*2 = 0.0805 calculated on *F*^2^*_hkl_* for all 2703 independent reflections with 2*θ* < 54.2° (*GOF* = 1.050, *R* = 0.0297 calculated on *F_hkl_* for 2300 reflections with *I* > 2σ(*I*)).

*Crystallographic data for compound***10g**: C_8_H_10_N_3_O^+^ClO_4_^−^ are triclinic, space group *P*-1: *a* = 11.2823(6) Å, *b* = 11.4350(6) Å, *c* = 11.6297(6) Å, α = 98.260(3)°, *β* = 118.986(2)°, γ = 110.895(2)°, *V* = 1130.45(11) Å^3^, *Z* = 4, M = 263.64, *d*_cryst_ = 1.549 g·cm^−3^. *wR*2 = 0.1358 calculated on *F*^2^*_hkl_* for all 4613 independent reflections with 2*θ* < 52.8° (*GOF* = 1.105, *R* = 0.0522 calculated on *F_hkl_* for 3871 reflections with *I* > 2σ(*I*)).

**3,5,7-Trimethyl-[1,2,5]oxadiazolo [2,3-*a*]pyrimidin-8-ium perchlorate** (**10a**). A mixture of 58% HClO_4_ (0.97 g, 5.6 mmol), CF_3_CO_2_H (3.5 mL), and 3-amino-4-methylfurazan (0.5 g, 5.0 mmol) was stirred for 5 min at rt, and pentane-2,4-dione (0.56 g, 5.6 mmol) was added. The mixture was stirred for 3 h at rt and then diluted with Et_2_O (20 mL). The precipitate was isolated by filtration, washed with EtOH (3 × 5 mL) and Et_2_O (2 × 5 mL), and dried under vacuum to give salt **10a** (80%) as a white solid: mp 213–214 °C dec [lit. [44] mp 195–197 °C (dec)]; IR (KBr) *ν* 3092, 2934, 1629, 1575, 1563, 1442, 1401, 1377, 1363, 1269, 1205, 1094 cm^−1^; ^1^H NMR (CD_3_CN) *δ* 2.84 (s, 3H), 2.95 (s, 3H), 3.05 (s, 3H), 8.06 (s, 1H); ^13^C NMR (CD_3_CN) *δ* 9.3, 16.2, 25.6, 122.7, 146.2, 151.1, 154.7, 177.9; ^15^N NMR (CD_3_CN) *δ* 268.8, 281.0, 389.2; ^35^Cl NMR (CD_3_CN) *δ* 1012.1. Anal. Calcd for C_8_H_10_ClN_3_O_5_ (263.63): C, 36.45; H, 3.82; N, 15.94. Found: C, 36.40; H, 3.79; N, 15.91.

**6-Chloro-3,5,7-trimethyl-[1,2,5]oxadiazolo [2,3-*a*]pyrimidin-8-ium perchlorate** (**10b**). Following the same procedure outlined above, 3-chloropentane-2,4-dione gave the desired product **10b** (85%) as a white solid: mp 180–181 °C dec; IR (KBr) *ν* 3026, 2934, 1605, 1552, 1395, 1266, 1095 cm^−1^; ^1^H NMR (CD_3_CN) *δ* 2.63 (s, 3H), 3.02 (s, 3H), 3.14 (s, 3H); ^13^C NMR (CD_3_CN) *δ* 8.9, 15.2, 25.1, 132.1, 143.4, 149.3, 154.5, 175.1. Anal. Calcd for C_8_H_9_Cl_2_N_3_O_5_ (298.08): C, 32.24; H, 3.04; N, 14.10. Found: C, 32.29; H, 3.07; N, 14.05.

**3-Chloromethyl-5,7-dimethyl-[1,2,5]oxadiazolo [2,3-*a*]pyrimidin-8-ium perchlorate** (**10e**). Following the same procedure outlined above, the reaction of 3-amino-4-chloromethylfurazan (**9b**) with pentane-2,4-dione gave the desired product **10e** (96%) as a white solid: mp 195–196 °C dec; IR (KBr) *ν* 3076, 3023, 2932, 1626, 1572, 1441, 1396, 1377, 1262, 1202, 1093, 779, 738, 625 cm^−1^; ^1^H NMR (CD_3_CN) *δ* 2.98 (s, 3H), 3.02 (s, 3H), 3.08 (s, 3H), 5.21 (s, 2H), 8.12 (s, 1H); ^13^C NMR (CD_3_CN) *δ* 15.3, 24.8, 31.8, 122.4, 143.7, 150.6, 153.0, 177.7. Anal. Calcd for C_8_H_9_Cl_2_N_3_O_5_ (298.08): C, 32.24; H, 3.04; N, 14.10. Found: C, 32.21; H, 3.02; N, 14.06.

**3-Azidomethyl-5,7-dimethyl-[1,2,5]oxadiazolo [2,3-*a*]pyrimidin-8-ium perchlorate** (**10f**). Following the same procedure, the reaction of 3-amino-4-azidomethylfurazan (**9c**) with pentane-2,4-dione gave the desired product **10e** (94%) as a white solid: mp 195–196 °C dec; IR (KBr) *ν* 3081, 2934, 2215, 2125, 1624, 1573, 1442, 1424, 1333, 1282, 1264, 1195, 1093 cm^−1^; ^1^H NMR (CD_3_CN) *δ* 2.93 (s, 3H), 3.05 (s, 3H), 3.08 (s, 3H), 5.04 (s, 2H), 8.08 (s, 1H); ^13^C NMR (CD_3_CN) *δ* 14.5, 23.9, 42.2, 121.4, 143.3, 149.7, 151.7, 176.7. Anal. Calcd for C_8_H_9_ClN_6_O_5_ (304.65): C, 31.54; H, 2.98; N, 27.59. Found: C, 31.60; H, 3.03; N, 27.56.

**3-Azido-5,7-dimethyl-[1,2,5]oxadiazolo [2,3-*a*]pyrimidin-8-ium perchlorate** (**10g**). Following the same procedure, the reaction of 3-amino-4-azidofurazan (**9d**) with pentane-2,4-dione gave the desired product **10g** (71%) as a white solid: mp 140–141 °C dec; IR (KBr) *ν* 3082, 2930, 2157, 1625, 1543, 1443, 1412, 1306, 1266, 1177, 1091 cm^−1^; ^1^H NMR (CD_3_CN) *δ* 2.96 (s, 3H), 3.05 (s, 3H), 8.13 (s, 1H); ^13^C NMR (CD_3_CN) *δ* 15.9, 25.6, 123.5, 140.8, 152.0, 153.0, 178.2. Anal. Calcd for C_7_H_7_ClN_6_O_5_ (290.62): C, 28.93; H, 2.43; N, 28.92. Found: C, 29.00; H, 2.47; N, 28.86.

## 4. Conclusions

A new group of furazan-based energetic materials, [1,2,5]oxadiazolo [2,3-*a*]pyrimidin-8-ium perchlorates bearing explosophoric groups, have been synthesized for the first time. The synthetic protocol does not require complex procedures, relying on the simple mixing of available reagents and the usual filtering of the desired product. All compounds were fully characterized by multinuclear NMR spectroscopy and X-ray crystal structure determinations. Initial safety testing (impact and friction sensitivity) and thermal stability measurements (DTA) were also carried out. These salts demonstrate an excellent burn rate and combustion behavior. Considering the simplicity of preparation and the inherent combination of properties, the [1,2,5]oxadiazolo [2,3-*a*]pyrimidin-8-ium backbone may be used as an effective building block in the creation of new energetic materials for various purposes.

## Data Availability

CCDC 2210557, 2210558 and 2210559 contain the supplementary crystallographic data for this paper. The data can be obtained free of charge from The Cambridge Crystallographic Data Centre via https://www.ccdc.cam.ac.uk/structures.

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
