# Peer review of "Energetic [1,2,5]oxadiazolo [2,3-a]pyrimidin-8-ium Perchlorates: Synthesis and Characterization"

_molecules, 2022, doi:10.3390/molecules27238443_

Round 1

Reviewer 1 Report

The publication "Synthesis and Characterization of [1,2,5]Oxadiazolo[2,3-a]pyrimidin-8-ium Perchlorates" generally is well written and fits perfectly with the theme of this special issue so i would recommend to "Accept after Minor Revisions".

In some cases it is hard to understand the meaning of some sentences,

for example lines 29-30 or 90-91, to name a few, so i would recommend to perform "english editing".

Throughout the publication, [2,3-a], [3,4-c] etc, i think [2,3-a], [3,4-c] we must use the italic font.

106 line, please correct  "trifluorimethyl" to "trifluoromethyl"

116 and 355 lines, please correct "3-амино-4-nitrofurazan", from russian language to english.

Scheme 2, the formulas of compounds 9b, c, d, e, f must be improved. Increase the size?

124 line, is it 1H-1H NOESY or 1D selective NOESY, please clarify.

In supplementary materials.

10a, 1D selective NOESY spectra looks poorly phased. As it is presented at the moment, the signals which are peak picked, looks like a noise, the real NOE's should be in opposite phase. Not somewhere in between.

Generally supporting information NMR part could be improved, for instance if 1H-15N HMBC spectral data is presented, i would show 1H NMR spectrum in f2 dimension. Some peak picking would make more clarity as well in 2D spectral data.

The chemical shifts of 1H, 13C, 15N and 35Cl are as expected for this class of compounds. No concerns.

Author Response

Comment 1: In some cases it is hard to understand the meaning of some sentences,for example lines 29-30 or 90-91, to name a few, so i would recommend to perform "english editing".

Response: English has been edited. In addition, some long sentences have been divided into two shorter ones for easier understanding.

Comment 2: Throughout the publication, [2,3-a], [3,4-c] etc, i think [2,3-a], [3,4-c] we must use the italic font.

Response: This is corrected.

Comment 3: 106 line, please correct  "trifluorimethyl" to "trifluoromethyl"

Response: This is corrected.

Comment 4: 116 and 355 lines, please correct "3-амино-4-nitrofurazan", from russian language to english.

Response: This is corrected.

Comment 5: Scheme 2, the formulas of compounds 9b, c, d, e, f must be improved. Increase the size?

Response: The size of formulas for compounds 9b-f in scheme 2 has been increased.

Comment 6: 124 line, is it 1H-1H NOESY or 1D selective NOESY, please clarify.

Response: This is corrected.

Comment 7: In supplementary materials. 10a, 1D selective NOESY spectra looks poorly phased. As it is presented at the moment, the signals which are peak picked, looks like a noise, the real NOE's should be in opposite phase. Not somewhere in between.

Generally supporting information NMR part could be improved, for instance if 1H-15N HMBC spectral data is presented, i would show 1H NMR spectrum in f2 dimension. Some peak picking would make more clarity as well in 2D spectral data.

The chemical shifts of 1H, 13C, 15N and 35Cl are as expected for this class of compounds. No concerns.

Response: In supplementary materials, the presentability of copies for NMR spectra has been improved. HRMS copies have also been added.

Reviewer 2 Report

This manuscript describes the synthesis and characterizations of oxadiazolo[2,3-a]pyrimidinium perchlorates bearing furazan based energetic scaffolds. They performed DTA and impact and friction sensitivity for these scaffolds. Also, the authors claims that these moieties might useful as key synthons for design of new energetic materials. Overall, this referee feels the manuscript fit to molecules criteria after improvement some concerns.

Comments

1.      Title and abstract need to be changed and revised with better manner.

2.      Scheme 1/2 caption and exp. Conditions should be mentioned in all cases with yields.

3.       NMR copies of all compounds [1H/13C] as well mass spectroscopic HRMS data should be included

4.      These are high-nitrogen content or nitrogen rich??

5.      Compound weights must be provided in all.

6.      There are some typos present in the text, please check
 over thoroughly and manuscript need to be improved with all details.

Author Response

Comment 1: Title and abstract need to be changed and revised with better manner.

Response: The title and abstract have been changed, for greater clarity and informativeness.

Comment 2: Scheme 1/2 caption and exp. Conditions should be mentioned in all cases with yields.

Response: The text next to Schemes describes the conditions and yields of all compounds. Duplicating this information on the Scheme is impractical, since it will clutter up an already saturated scheme. Note that another reviewer recommended increasing the size of compounds 9b-f, which made the circuit even more saturated.

Comment 3. NMR copies of all compounds [1H/13C] as well mass spectroscopic HRMS data should be included

Response: Copies of all the spectra are in the Supporting Materials.

Comment 4: These are high-nitrogen content or nitrogen rich??

Response: These terms are equivalent. In modern literature, both are widely used.

Comment 5: Compound weights must be provided in all.

Response: Compound weights are presented in Table 1 and 4. They are also given in the experimental part.

Comment 6: There are some typos present in the text, please check over thoroughly and manuscript need to be improved with all details.

Response: English has been edited. In addition, some long sentences have been divided into two shorter ones for easier understanding.

A number of minor edits have also been made to the manuscript, which are highlighted with a marker.

Round 2

Reviewer 2 Report

The revised version reasonably fine and it can be accepted for publication.

with best wishes